# Effects of the Replacement of Dietary Fish Meal with Poultry By-Product Meal on Growth and Intestinal Health of Chinese Soft-Shelled Turtle (*Pelodiscus sinensis*)

**DOI:** 10.3390/ani13050865

**Published:** 2023-02-27

**Authors:** Zongsheng Qiu, Qiyou Xu, Dazhang Xie, Jiantao Zhao, Fernando Y. Yamamoto, Hong Xu, Jianhua Zhao

**Affiliations:** 1Zhejiang Provincial Key Laboratory of Aquatic Resources Conservation and Development, School of Life Science, Huzhou University, 759 Erhuan Road (E), Huzhou 313000, China; 2Zhejiang Jindadi Biotechnology Co., Ltd., Shaoxing 311800, China; 3Thad Cochran National Warmwater Aquaculture Center, Mississippi Agriculture and Forestry Experiment Station, Mississippi State University, Starkville, MS 39762, USA; 4Department of Wildlife, Fisheries and Aquaculture, Mississippi State University, Starkville, MS 39762, USA

**Keywords:** poultry by-product meal, growth, immunity, antioxidation, intestinal health, *Pelodiscus sinensis*

## Abstract

**Simple Summary:**

Fish meal is widely used in aquaculture feed due to its high protein content, balanced amino acid composition, and low anti-nutritional factors. However, with the development of intensive aquaculture, fish meal resources are in short supply and prices are rising. In order to ensure the sustainable development of aquaculture, it is very important to find a low-cost, high-quality protein source to replace fish meal. In this study, substituting fish meal of Chinese soft-shelled turtle (*Pelodiscus sinensis*) diets with poultry by-product meal was carried out. The results showed that replacing fish meal with 15% poultry by-product meal had no adverse effect on the growth of Chinese soft-shelled turtle (*Pelodiscus sinensis*), and the 10% poultry by-product meal was beneficial to the growth performance, immunity, and digestive ability. Therefore, it is feasible to use poultry by-product meal as a protein source to replace fish meal in Chinese soft-shelled turtle (*Pelodiscus sinensis*) feed.

**Abstract:**

To investigate the effect of poultry by-product meal (PBM) replacing fish meal on the growth and intestinal health of Chinese soft-shelled turtle *(Pelodiscus sinensis)*. Four experimental diets were prepared. Fish meal was replaced by 0 (control group, PBM0), 5% (PBM5), 10% (PBM10), and 15% (PBM15) PBM. Compared to the control group, final body weight, weight gain, and specific growth rate were significantly increased, while feed conversion rate decreased significantly in the PBM10 group (*p* < 0.05). The PBM15 group significantly increased the moisture content and significantly decreased the ash content of the turtles (*p* < 0.05). The PBM5 and PBM15 groups significantly decreased the whole-body crude lipid (*p* < 0.05). The serum glucose content increased significantly in the PBM10 group (*p* < 0.05). The liver malonaldehyde content significantly decreased in the PBM5 group and in the PBM10 group (*p* < 0.05). Liver glutamic-oxalacetic transaminase and intestinal pepsin activity were increased significantly in the PBM15 group (*p* < 0.05). The expression of the intestinal interleukin 10 (*IL-10*) gene was significantly down-regulated in the PBM10 group and the PBM15 group (*p* < 0.05), the expression of the intestinal interferon-*γ* (*IFN*-*γ*), interleukin-8 (*IL-8*), and liver toll-like receptor 4 (*TLR4*) and toll-like receptor 5 (*TLR5*) genes were significantly up-regulated in the PBM5 group (*p* < 0.05). In summary, poultry by-product meal can be used as a protein source to replace fish meal in turtle feed. Based on quadratic regression analysis, the optimal replacement ratio is 7.39%.

## 1. Introduction

Fish meal has always been regarded as the highest quality protein material for aquaculture feed due to its high protein content, good palatability, low anti-nutritional factors, and easy digestion [1]. In recent years, fish meal production has decreased. With aquaculture expansion, the demand for fish meal has also increased [2], and the price has increased, seriously restricting the sustainable development of aquaculture [3]. The search for protein resources that can replace fish meal has become an inevitable trend in the development of aquaculture feed. Currently, the substitution of fish meal can be roughly divided into three directions: animal protein source [4,5], plant protein source [6,7], and mixed protein source [8,9].

The type of protein in feed plays an important role in the growth and development of fish. However, the presence of a large number of anti-nutritional factors in plant protein feed [10], amino acid imbalance, and low utilization dramatically limits the use of plant protein in aquatic animal feed [11]. Poultry by-product meal (PBM) is a new type of animal protein feed raw material processed from the remaining part of the breast meat of chickens removed from the shelf and white-striped chickens. The overall nutritional index of poultry by-product meal is comparable to that of fish meal. The content of protein, fat, and some vitamins is slightly higher than that of high-quality fish meal, and the freshness is better controlled than fish meal [12]. In recent years, it has been widely used in the United States and Southeast Asia. Studies have shown that an appropriate amount of PBM instead of fish meal has positive results in feeds such as white shrimp (*Litopenaeus vannamei*) [13], snakehead (*Channa argus*) [14], black sea bream (*Acanthoparus schlegelii*) [15], and nile tilapia (*Oreochromis niloticus*) [16]. PBM has a high content of saturated fatty acids and ash, affecting fish digestion and absorption [17]. Meanwhile, the high proportion of replacement of fish meal will affect the growth performance of aquatic animals [18]. Goda et al. [19] showed that replacing 75% and 100% of fish meal with PBM affected the growth performance of African catfish (*Clarias gariepinus*). Jr et al. [20] found that replacing fish meal with PBM led to a decrease in the growth performance of pompano (*Trachinotus carolinus*). Turkey et al. [21] found that the optimal replacement ratio of fish meal by PBM in a turbot diet was 25%. The effects of PBM replacing fish meal on the growth performance of aquatic animals were different in different species.

The Chinese soft-shelled turtle is widely cultured because of its high nutritional value and high breeding efficiency. However, the high dependence on fish meal of the Chinese soft-shelled turtle has seriously restricted the development of its breeding industry. It is urgent to develop a new protein source to replace fish meal, so this experiment intended to use PBM instead of fish meal on the growth, immunity, and intestinal health of Chinese soft-shelled turtles and provided theoretical support for low fish meal feed.

## 2. Materials and Methods

### 2.1. Experimental Animals, Diets, and Feeding Trial

The experiment was carried out in the greenhouse breeding system of Zhuji Jindadi Agriculture Co., Ltd., Shaoxing City, Zhejiang Province. 480 turtles with an average weight of 3.48 ± 0.01 g were randomly distributed in 16 cages (0.6 m × 0.6 m × 0.6 m) with 30 tails per cage. Each dietary group had four replicates, and 30 turtles were stocked in each replicate cage. The turtles were fed three times a day (6:00, 12:00, and 18:00 h) and the feeding amount was set at 4% of the body weight. The chemical composition of fish meal and the poultry by-product meal used in the present study is shown in Appendix A. PBM was used to replace fish meal in a ratio of 0 (PBM0 (control group), 5% (PBM5), 10% (PBM10), and 15% (PBM15), and the basic feed is shown in Appendix A. The amino acid composition of each group is shown in Appendix A. All ingredients were finely ground and thoroughly mixed in a blender. The 2.2 mm expanded particles were made by an expanded granulator and dried by an air dryer.

The feeding trial lasted eight weeks. During the experiment, the water temperature was 30 ± 1 °C, the dissolved oxygen was kept at more than 5 mg/L, the ammonia nitrogen was less than 0.8 mg/L, and the nitrite was less than 0.05 mg/L.

The turtles in each cage were weighed at the start (initial body weight) and end (final body weight) of the 56−day feeding experiment. The survival rate, weight gain, and feed coefficient of each treatment group were recorded and analyzed. These parameters were calculated as follows:Survival rate (%) = (Nf/Ni) × 100; 
Weight gain (WG, %) = (Wf − Wi)/Wi × 100; 
SGR (specific growth rate, %/day) = ((ln Wf – ln Wi)/days) × 100; 
Feed conversion ratio (FCR) = Feed consumed (g)/(Wf − Wi); 
where W_f_ and W_i_ are the initial and final body weight, N_f_ and N_i_ are the final and initial numbers of turtles.

### 2.2. Sample Collection

At the end of the feeding trial, three turtles were randomly selected from each replicate and used for whole-body composition analysis [22]. The moisture content was determined by the 105 °C constant temperature drying method. The crude protein content was determined by the Kjeldahl nitrometer. The crude lipid was determined by soxhlet extraction. Ash content was determined by Muffle furnace combustion at 550 °C. Three turtles per cage were decapitated, and blood was collected in a 2 mL centrifuge tube, placed at 4 °C for 6 h, and centrifuged at 3000 rpm for 10 min to obtain serum samples. The turtle was then dissected, and the liver and intestine were separated. The tissue was mixed with a 0.9% sodium chloride solution at a ratio of 1:9 (mass/volume), homogenized with a homogenizer, then centrifuged at 2500 rpm for 10 min, and the supernatant was stored in a refrigerator at −80 °C for further analysis. Finally, the collected samples were mixed in pairs, and 6 samples were obtained from each treatment group. The midguts were taken and placed in Bouin’s fixative solution to make histological sections.

### 2.3. Biochemical Analysis

Serum total cholesterol (T−CHO), blood urea nitrogen (BUN), glucose (GLU), triglyceride (TG), high−density lipoprotein cholesterol (HDL−C), and low−density lipoprotein cholesterol (LDL−C) were measured. The contents of superoxide dismutase (SOD), reduced glutathione (GSH), catalase (CAT), malondialdehyde (MDA), and aspartate aminotransferase (GOT) in the liver were determined. The activities of intestinal trypsin, pepsin, lipase, and amylase were determined. Kits produced by the Nanjing Jiancheng Institute of Bioengineering were used for determination and analysis.

### 2.4. Intestinal Histological Analysis

The midgut tissue was fixed in Bouin’s fixative solution for 48 h, which was dehydrated with ethanol, transparentized with xylene, embedded in paraffin, sectioned, and stained with hematoxylin–eosin (HE). And finally, the morphological characteristics of the intestinal villi were observed under a light microscope; images were taken; and villi height, width, and thickness of the muscular layer were measured with a micrometer ruler.

### 2.5. Analysis of Intestinal and Liver Gene Expression

Total RNA was extracted from turtle intestinal and liver tissue samples according to the RNA rapid extraction kit (RN28−EASYspin Plus), and cDNA was synthesized by the reverse transcription kit (MonScript TM 5 × RTIIII All−in−One Mix, Mona, Suzhou, China). The antibody dye quantitative PCR master mix (Mon Amp TM SYBR^®^ Green q PCR Mix) was used to perform RT−qPCR on the real−time quantitative PCR instrument (ABI 7500), and specific operation steps were carried out according to the instructions. The reaction system was 20 μL, including 10 μL MonAmp TM SYBR^®^ Green qPCR Mix (Mona, Suzhou, China), 0.4 μL forward and reverse primers (10 μmol/L), 1.5 μL template cDNA, 7.7 μL nuclease−free water. The thermal cycle program of qRT−PCR was as follows: 95 °C for 5 min, then 40 cycles at 95 °C for 10 s, 60 °C for 30 s, and 72 °C for 30 s. The melting curve was drawn to determine the correctness of the amplification product. By making a gradient dilution concentration standard curve, it was found that the amplification efficiency of the target gene and the internal reference gene was close to 1 [23], which was consistent with the target gene 2^−ΔΔCt^ method. The primer sequences of genes related to intestinal and liver health−related genes in Chinese soft-shelled turtles are shown in Appendix A. 

### 2.6. Statistical Analysis

The experimental data were analyzed by one−way analysis of variance (ANOVA) using SPSS 26.0 software, and the data for each group were compared using the Turkey method for multiple comparisons, and p < 0.05 was considered a significant difference. Experimental data are expressed as the mean ± standard error (mean ± SE).

## 3. Results

### 3.1. Growth Performance

There was no significant difference in IBW and SR among all groups (*p* > 0.05). Compared to the control group, the PBM10 group significantly increased FBW, WG, and SGR (*p* < 0.05) while significantly decreased turtle FCR (*p* < 0.05) (Table 1). Based on the second−regression analysis model between WG and dietary PBM levels, the PBM optimal substitution level for fish meal was 7.39% (Figure 1).

### 3.2. whole-body Composition

Compared to the control group, the PBM15 group significantly increased the moisture content and significantly decreased the ash content of the turtles (*p* < 0.05). The PBM5 and PBM15 groups significantly decreased the whole-body crude lipid (*p* < 0.05). The dietary groups PBM5, PBM10, and PBM15 had no significant effect on the crude protein of the turtles (*p* > 0.05) (Table 2).

### 3.3. Serum Biochemical Indicators

There were no significant differences in serum levels of T−CHO, BUN, TG, HDL−C, and LDL−C among all groups (*p* > 0.05). Compared to the control group, the PBM10 group significantly increased serum GLU content (*p* < 0.05) (Figure 2).

### 3.4. Antioxidant Parameters of the Liver

There were no significant differences in liver SOD, CAT activity, and GSH content among all groups (*p* > 0.05). Compared to the control group, the PBM5 group and the PBM10 group significantly decreased the liver MDA content (*p* < 0.05), and the PBM15 group significantly increased liver GOT activity (*p* < 0.05) (Figure 3).

### 3.5. Intestinal Digestive Enzyme Activity

There were no significant differences in trypsin, lipase, and amylase activities among all groups (*p* > 0.05). Compared to the control group, the PBM5 group and the PBM10 group significantly increased intestinal pepsin activity (*p* < 0.05) (Figure 4).

### 3.6. Histological Study of the Intestine

There were no significant differences in intestinal villi length and width among all groups (*p* > 0.05). As the proportion of PBM that replaced fish meal was increased, the thickness of the muscle layer gradually was decreased, and compared to the PBM5 group, the thickness of the muscle layer was significantly decreased in the PBM15 group (*p* < 0.05) (Table 3, Figure 5).

### 3.7. Expression of Intestinal and Liver Genes

There were no significant differences in the expression levels of intestinal *IL−1β* and *IL−15* genes among all groups (*p* > 0.05). Compared to the control group, the PBM10 and PBM15 groups significantly down-regulated the intestinal *IL-10* gene expression (*p* < 0.05), and the PBM5 group significantly up-regulated the intestinal *IFN-γ* and *IL-8* gene expressions (*p* < 0.05) (Figure 6). There were no significant differences in the expression levels of *IL−1β*, *IGF−1*, and *TLR8* in the liver (*p* > 0.05). The expressions of the *TLR4* and *TLR5* genes in the liver were significantly up-regulated in the PBM5 group (*p* < 0.05) (Figure 7).

## 4. Discussion

This study shows that PBM can be used as a protein source instead of fish meal in Chinese soft-shelled turtle feed. The turtle had a good tolerance to PBM, and the growth performance and feed utilization rate were the best when the fish meal replacement ratio was 7.39%. Substituting fish meal with PBM in the diet can improve turtle growth performance, which is consistent with previous findings in various species, including rainbow trout (*Oncorhynchus mykiss*) [24], flounder (*Paralichthys olivaceus*) [25], and black sea bream (*Acanthopagrus schlegelii*) [3]. These studies have shown that PBM meets the nutritional and energy needs of aquatic animals and promotes their growth. According to Sayed et al. [26] and Wang et al. [27], increasing the proportion of PBM instead of fish meal had a negative impact on the growth performance of aquatic animals. In this study, the replacement of 15% fish meal with PBM did not affect turtle growth performance but showed a decreasing trend, which is consistent with the results of Ji et al. [28]. This result may be because turtles are typical carnivores and prefer diets with a higher proportion of fish meal [29].

Poultry by-product meal altered the body composition of turtles. When 15% fish meal was replaced by poultry by-product meal, the moisture content of Pelodiscus sinensis increased and the ash content decreased. When 5% and 15% fish meal were replaced by poultry by-product meal, the crude fat content of *p. sinensis* decreased. The same poultry by-product meal also affects the body composition of turbot [17,21]. Studies have shown that the replacement of fish meal with PBM has no significant effect on the body composition of crucian carp [30] and tilapia [31], but Zhou et al. found that the replacement of fish meal with PBM significantly decreased the protein content of juvenile cobia (*Rachycentron canadum*) [32]. This result may be due to different species, and the replacement of fish meal with PBM has other effects on the whole-body composition, and the specific mechanism needs to be further explored.

Serum indicators reflect the health status, physiological function, and nutritional level of the body [33]. Serum levels of T−CHO and TG reflect the body’s lipid metabolism [34], and serum levels of HDL−C and LDL−C represent the breakdown and transport of body lipids [35]. The serum content of BUN reflects the protein metabolism and amino acid balance of the body [36]. In this experiment, the replacement of fish meal by PBM did not affect the serum levels of T−CHO, TG, BUN, HDL−C, and LDL−C of turtles, which is consistent with the results of Sugita et al. [37] in milkfish (*Chanos chanos*). The serum GLU content of the PBM10 group increased, indicating that the body’s energy level was higher [38], and the growth rate was accelerated, indicating that the replacement of fish meal with PBM could affect the energy metabolism of the Chinese soft-shelled turtle.

As an important immune organ of aquatic animals, the liver has the functions of immune metabolism, detoxification, and anti−oxidation [39]. SOD, CAT, and GSH play important roles in maintaining the balance of oxidation and anti−oxidation in the body and avoiding oxidative damage [40]. MDA is the main peroxidation−decomposing substance in the body, whose concentration directly reflects the degree of peroxidation in the body and the degree of damage to cells [41]. In this study, the replacement of fish meal with PBM did not affect SOD, CAT, or GSH in the liver of turtles, which was consistent with the results of Zhou et al. [32]. Replacement of 5% and 10% fish meal with PBM could reduce the content of MDA in the liver, indicating that the appropriate proportion of PBM instead of fish meal would be beneficial to the liver health of Chinese soft-shelled turtles. GOT is an important amino acid metabolic enzyme in the liver of aquatic animals [42]. The increase in GOT activity indicates that proteins are used to enhance energy metabolism [43]. In this study, GOT activity in the liver increased with increased levels of PBM replacement, which is consistent with the results of Lu et al. [44] on rainbow trout. This result may be due to the increase in the amino acid imbalance in the feed with increasing PBM, which leads to an improvement in amino acid metabolism in the liver of turtles and an improvement in GOT activity to adapt to the increase in PBM content in the feed.

Enhancement of digestive enzyme activity can promote food digestion and nutrient absorption in the body, thereby promoting fish growth. In this study, different proportions of PBM replacing fish meal had no significant effect on trypsin, lipase, and amylase activities. Pepsin activity first increased and then decreased with increasing PBM replacement ratios, which is consistent with the results in sea bass [45]. The above studies showed that properly replacing fish meal with PBM could increase pepsin activity in the intestinal tract of turtles and then improve growth performance. Intestinal villus length, width, and muscle thickness can help predict the absorption and digestion mechanisms in aquatic animals [46]. In this study, the replacement of PBM with fish meal did not affect the length and width of the intestinal villi. But with the increase in the PBM replacement ratio, the thickness of the muscle layer gradually decreased. Muscle thickness can reflect the contractile capacity of the intestine [47]. The results showed that when the high proportion of fish meal was replaced by PBM, it had a negative impact on the intestinal contractile capacity, which in turn affected the intestinal absorption capacity of turtles.

Aquatic animals can regulate the immune response by enhancing or inhibiting the production of cytokines [48]. Cytokines in aquatic animals can be divided into anti−inflammatory (such as *IL-10* and *TGF−β1*) and pro−inflammatory (such as *TNF−α*, *IL−1β*, *IL−15*, and *IL-8*) [49]. Inflammatory injury can be measured by the expression levels of pro−inflammatory factors and anti−inflammatory factors [50]. Yang et al. [18] found that replacing 25%, 50%, 75%, and 100% fish meal with PBM did not show obvious enteritis symptoms in red swamp crayfish (*Procambarus clarkii*). Basilio et al. [51] found that PBM completely replaced plant protein sources in gilthead seabream (*Sparus aurata*) feed, and the response rate to inflammation was decreased. In this study, we found that the expression of the *IL-10* gene in the intestinal tract of turtles decreased with an increasing proportion of PBM. *IFN-γ* and *IL-8* genes were significantly up-regulated when the proportion of PBM replacing fish meal was 5%. However, when PBM replaced fish meal at 10% and 15%, the expressions of the inflammatory factors *IFN-γ* and *IL-8* genes were not up-regulated. The specific mechanism needs to be explored further. These results indicate that PBM does not have a negative effect on the intestinal health of turtles when replacing 10% and 15% of fish meal. toll-like receptors are mediators that connect specific and non−specific immunity and can control cytokine synthesis and release to regulate inflammatory responses [52]. The *TLR4*, *TLR5,* and *TLR8* molecules belong to MyD88−dependent signaling pathways in TLRs, and their expression levels can reflect the immune system’s strength [53]. In this study, PBM replacing 5% fish meal could up−regulate the expression of the *TLR4* and *TLR5* genes in the liver. This study showed that PBM might activate turtle liver cells through the *TLR4* and *TLR5* pathways, cause an immune response, and improve immunity.

## 5. Conclusions

In summary, this study showed that replacing fish meal with poultry by-product meal did not affect growth performance, feed utilization, immunity, antioxidant capacity, or intestinal health of turtles. Based on the quadratic regression analysis between WG and the level of poultry by-product meal in feed, the optimal ratio of poultry by-product meal replacing fish meal was 7.39%.

## Figures and Tables

**Figure 1 animals-13-00865-f001:**
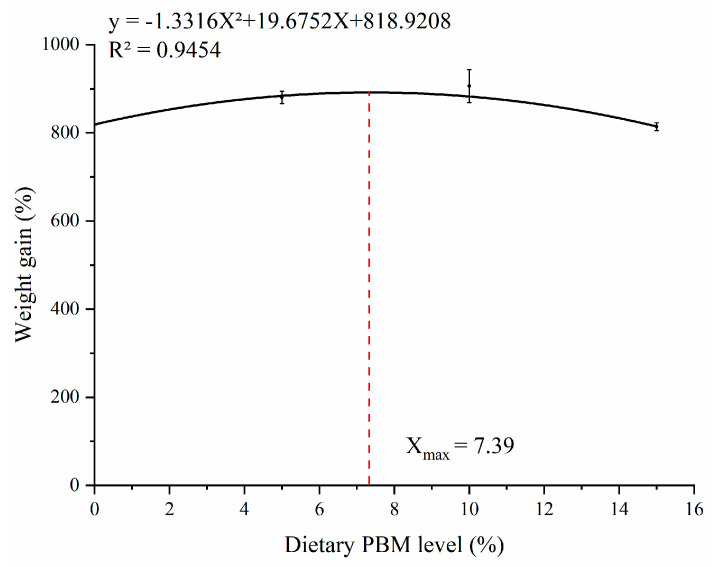
The relationship between WG and dietary PBM replacement level in Chinese soft-shelled turtle.

**Figure 2 animals-13-00865-f002:**
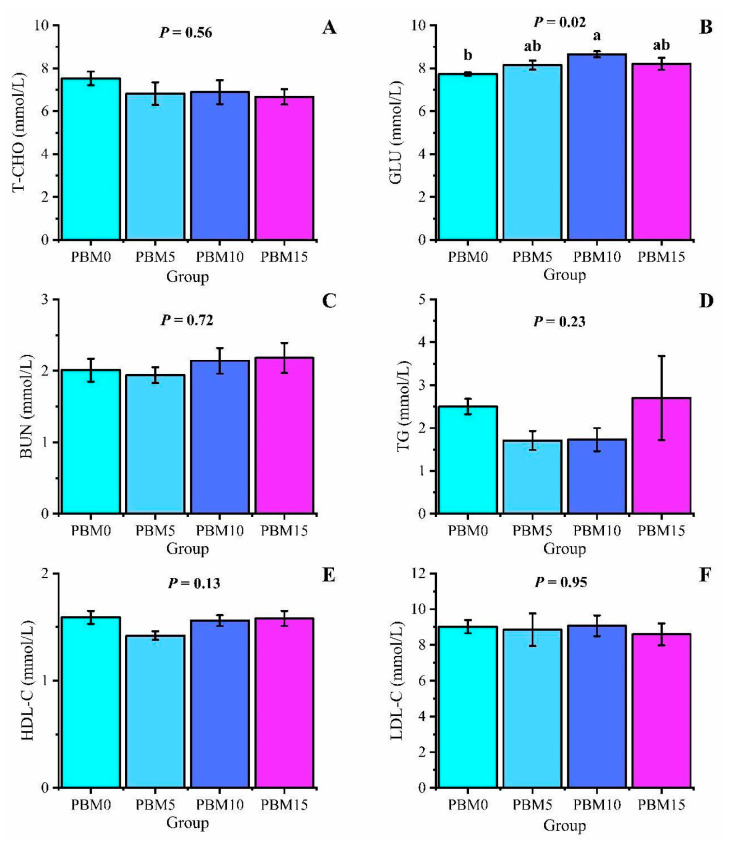
Effect of dietary replacement of fish meal with poultry by-product meal on the serum biochemical indicators of Chinese soft-shelled turtle. (**A**) T−CHO, total cholesterol; (**B**) GLU, glucose; (**C**) BUN, blood urea nitrogen; (**D**) TG, triglyceride; (**E**) HDL−C, high−density lipoprotein cholesterol; (**F**) LDL−C, low−density lipoprotein cholesterol; Error bars was represented the mean ± standard error (n = 6). Different letters indicated significant differences between groups (*p* < 0.05).

**Figure 3 animals-13-00865-f003:**
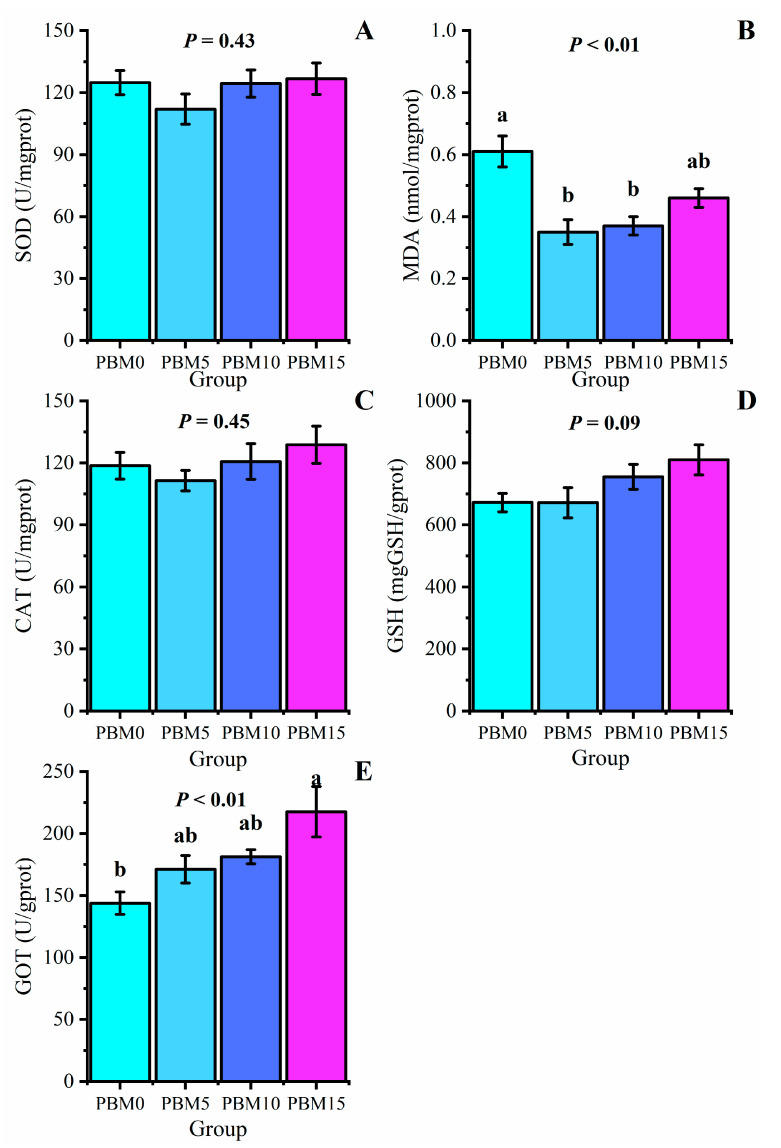
Effect of dietary replacement of fish meal with poultry by-product meal on the liver antioxidant of Chinese soft-shelled turtle. (**A**) Sod, superoxide dismutase; (**B**) MDA, malonaldehyde; (**C**) CAT, catalase; (**D**) GSH, glutathione; (**E**) GOT, glutamic-oxalacetic transaminase. The error bars represented the mean ± standard error (n = 6). Different letters indicated significant differences between groups (*p* < 0.05).

**Figure 4 animals-13-00865-f004:**
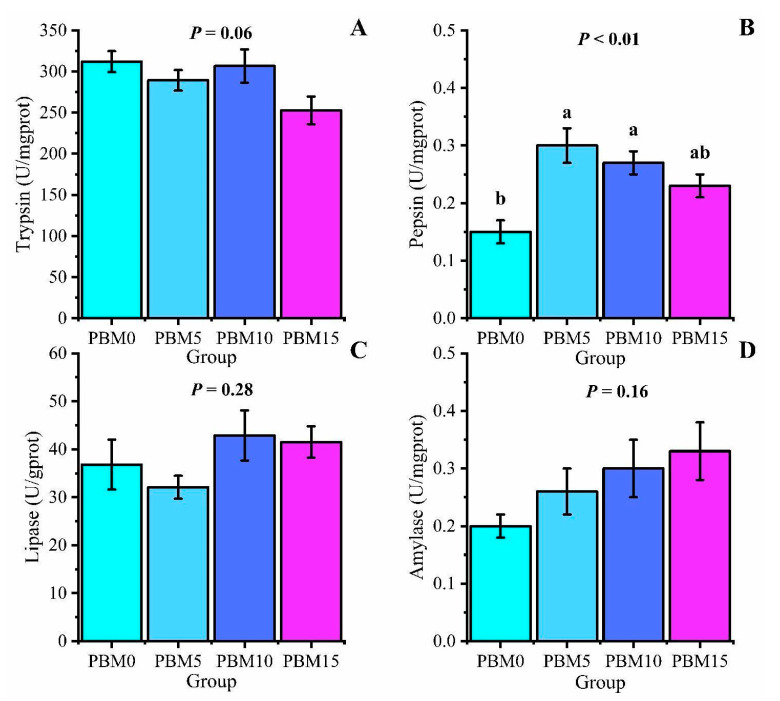
Effect of dietary replacement of fish meal with poultry by-product meal on the intestinal digestive enzyme activity of Chinese soft-shelled turtle. (**A**) Trypsin; (**B**) pepsin; (**C**) lipase; (**D**) amylase; Error bars was represented the mean ± standard error (n = 6). Different letters indicated significant differences between groups (*p* < 0.05).

**Figure 5 animals-13-00865-f005:**
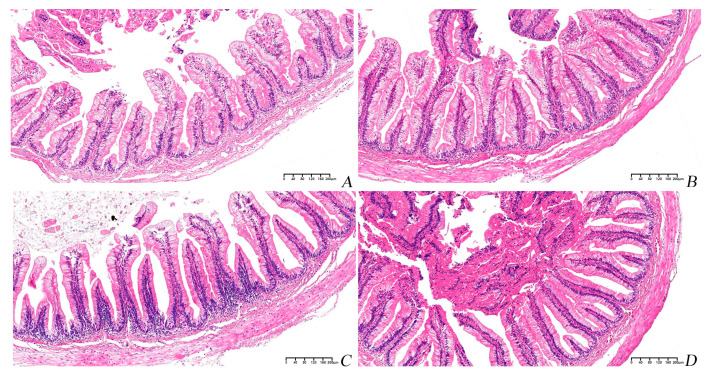
Histological sections of the intestines of Chinese soft-shelled turtle. Magnification times are 100×. (**A**) Turtle fed with Con diet. (**B**) Turtle fed with PBM5 diet. (**C**) Turtle fed with PBM10 diet. (**D**) Turtle fed with PBM15 diet. The scale bar was 200 μm.

**Figure 6 animals-13-00865-f006:**
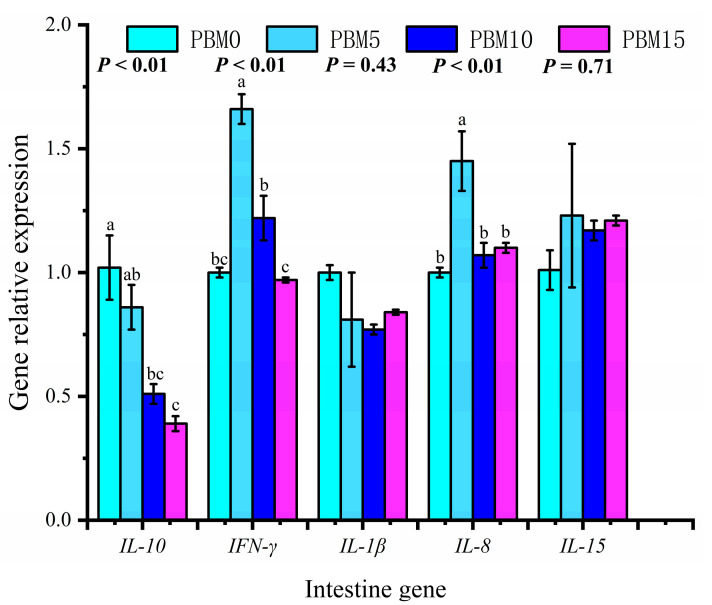
Effects of dietary replacement of fish meal with poultry by-product meal on the intestinal gene expression of Chinese soft-shelled turtle. Error bars was represented the mean ± standard error (n = 6). Different letters indicated significant differences between groups (*p* < 0.05).

**Figure 7 animals-13-00865-f007:**
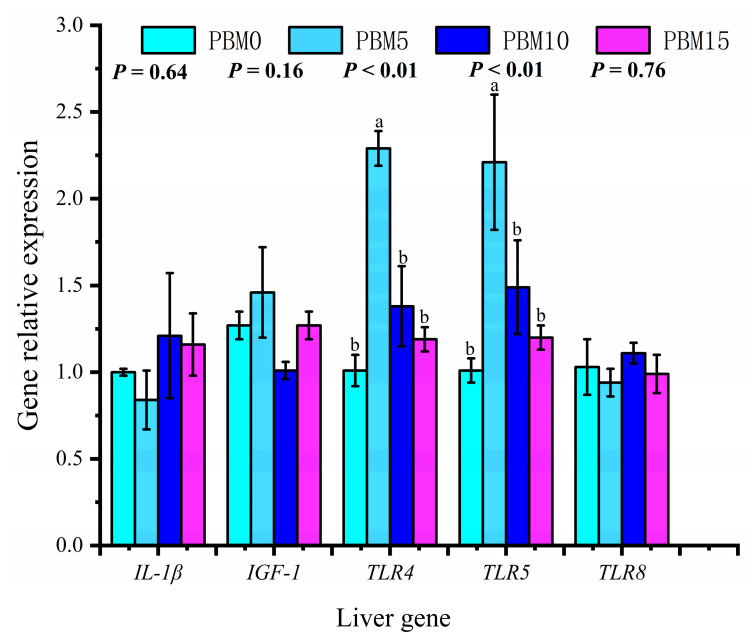
Effects of dietary replacement of fish meal with poultry by-product meal on the liver gene expression of Chinese soft-shelled turtle. Error bars was represented the mean ± standard error (n = 6). Different letters indicated significant differences between groups (*p* < 0.05).

**Table 1 animals-13-00865-t001:** Effect of dietary replacement of fish meal with poultry by-product meal on the growth performance of Chinese soft-shelled turtle.

Groups	IBW (g)	FBW (g)	SR (%)	WG (%)	SGR (%/d)	FCR
PBM0	3.47 ± 0.00	31.89 ± 0.06 ^b^	91.67 ± 2.15	818.93 ± 1.27 ^b^	3.96 ± 0.00 ^b^	0.81 ± 0.02 ^ab^
PBM5	3.48 ± 0.00	34.10 ± 0.48 ^ab^	92.50 ± 2.85	880.71 ± 13.90 ^ab^	4.08 ± 0.03 ^ab^	0.80 ± 0.01 ^b^
PBM10	3.47 ± 0.00	34.93 ± 1.30 ^a^	91.67 ± 3.19	906.13 ± 37.20 ^a^	4.12 ± 0.07 ^a^	0.77 ± 0.03 ^b^
PBM15	3.48 ± 0.00	31.81 ± 0.30 ^b^	92.50 ± 2.85	813.99 ± 8.92 ^b^	3.95 ± 0.02 ^b^	0.88 ± 0.02 ^a^
*p*−Value	0.31	0.02	0.99	0.02	0.02	0.02

Note: IBW: initial body weight; FBW: final body weight; WG: weight gain; SR: survival rate; FCR: feed conversion rate; SGR: specific growth rate. Values are presented as means ± standard error (n = 4). Mean values in the same row with different superscripts are significant different (*p* < 0.05).

**Table 2 animals-13-00865-t002:** Effect of dietary replacement of fish meal with poultry by-product meal on the whole-body proximate composition of Chinese soft-shelled turtle.

Groups	Moisture (%)	Crude Protein (%)	Crude Lipid (%)	Ash (%)
PBM0	69.46 ± 0.76 ^bc^	19.59 ± 0.58 ^ab^	4.24 ± 0.15 ^a^	5.24 ± 0.08 ^a^
PBM5	71.39 ± 0.61 ^ab^	18.25 ± 0.49 ^ab^	3.54 ± 0.10 ^b^	4.80 ± 0.09 ^ab^
PBM10	68.67 ± 0.69 ^c^	20.46 ± 0.60 ^a^	4.16 ± 0.17 ^a^	5.10 ± 0.07 ^a^
PBM15	73.90 ± 0.43 ^a^	17.55 ± 0.51 ^b^	3.42 ± 0.10 ^b^	4.53 ± 0.20 ^b^
*p*−Value	<0.01	0.01	<0.01	<0.01

Note: Values are presented as means ± standard error (n = 4). Mean values in the same row with different superscripts are significant different (*p* < 0.05).

**Table 3 animals-13-00865-t003:** Effects of dietary replacement of fish meal with poultry by-product meal on the intestinal villus height, villus width, and muscle layer thickness of Chinese soft-shelled turtle.

Groups	Villus Height (μm)	Villus Width (μm)	Muscle Layer Thickness (μm)
PBM0	373.62 ± 9.11	109.21 ± 4.87	72.98 ± 2.58 ^ab^
PBM5	414.65 ± 18.98	111.64 ± 6.17	80.39 ± 8.99 ^a^
PBM10	409.46 ± 16.65	110.40 ± 6.62	71.84 ± 5.36 ^ab^
PBM15	354.38 ± 19.19	97.52 ± 6.95	55.17 ± 2.92 ^b^
*p*−Value	0.05	0.37	0.03

Note: Values are presented as means ± standard error (n = 6). Mean values in the same row with different superscripts are significant different (*p* < 0.05).

## Data Availability

All data presented this study are available from the corresponding author, upon responsible request.

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
