# Peer review of "Effects of the Replacement of Dietary Fish Meal with Poultry By-Product Meal on Growth and Intestinal Health of Chinese Soft-Shelled Turtle (Pelodiscus sinensis)"

_animals, 2023, doi:10.3390/ani13050865_

Round 1

Reviewer 1 Report

The authors have documented that the poultry by-product meal (PBM) could affect turtles  growth, body composition,immunity, digestion, liver health, and intestinal health. Its a piece of nice work. However, in this manuscript, there are some points needed to be verified or improved before it is accepted for a publication in the Journal of Animals, MPDI.

Major:

1. The title is not appropriate, the authors should change it.

2. The abstract should be rewritten. The details on experiment setting can be simplified or deleted in this section. The abbreviations of can not be firstly appear in abstract.   

3. The authors should give more details and accurate description for the samples collections, for example, how many individuals of each group were examined for the grow performance, and other examinations respectively, in some tables, n=4, the other n=6, whats the meaning?

4. The Table 1 and Table 2 are not the results and provided limited information, these two tables should be provided as supplemental files.

Reviewer 2 Report

1. Full compositions of FM and PBM should be given to support diet composition 

2. Amino acid profiles of components and diets should be given. 

3 +/- or SD values are a nice addition, but there is a lack of p values among graphs. 

4. My general reception of the manuscript is very positive 

Reviewer 3 Report

attached in review report

Reviewer 4 Report

The manuscript with ID (animals-2201311) by Qiu and coauthors has evaluated the potential and possibilities of dietary replacing of fish meal with poultry by-product meal on the growth performance, serum immunity, antioxidant status, and intestinal health of Chinese Soft-shelled turtle. In general, this manuscript is interesting; however, several MAJOR revisions are present. The authors should revise them carefully before the manuscript considered for publication in Animals. Authors should prepare a suitable point-by-point to the comments and criticisms raised by the anonymous reviewer.

Q1. Please see highlights present in the PDF file combined with my reviews.

Q2. Revise the references as present in the PDF file.

Q3. Table 2: You should add the following points beside each gene

1. NCBI GenBank Accession number

2. Tm and annealing temperatures

3. Product size

4. References from which you got these primers

5. Gene efficiency %

6. You also should write the names of the used genes below the table

7. All gene abbreviations should be written italic in this position and throughout the whole manuscript

These data are very important for the readers in order to check the possible replication of your study.

Q4. Table 4: Why there were mortalities in the control group?

Q5. I need you to separate the results of the gene expression in intestine and gene expression in liver. Each one should be in a separate figure.

Q6: How the authors evaluated the intestinal histomorphometric characteristics of the intestines without providing their microscopic photos? You should provide them in the revised manuscript.

Q7. You should make polynomial regression analysis on the measured parameters in order to determine the best value of the poultry by products that can replace FM efficiently with no effects on the growth as you said in Line 309.

Round 2

Reviewer 1 Report

The authors have responded to most of questions raised by reviewers and the manuscript is acceptable for a publication now.

Author Response

-

Reviewer 4 Report

The authors have properly addressed my comments in the modified manuscript.

Author Response

-